# Aberrant Auditory and Visual Memory Development of Children with Upper Limb Motor Disorders

**DOI:** 10.3390/brainsci11121650

**Published:** 2021-12-15

**Authors:** Maria Koriakina, Olga Agranovich, Ekaterina Petrova, Dzerassa Kadieva, Grigory Kopytin, Evgenia Ermolovich, Olesya Moiseenko, Margarita Alekseeva, Dimitri Bredikhin, Beatriz Bermúdez-Margaretto, Ioannis Ntoumanis, Anna N. Shestakova, Iiro P. Jääskeläinen, Evgeny Blagovechtchenski

**Affiliations:** 1Federal State Budgetary Institution the Turner Scientific Research Institute for Children’s Orthopedics under the Ministry of Health of the Russian Federation, 196603 St. Petersburg, Russia; olga_agranovich@yahoo.com (O.A.); pet_kitten@mail.ru (E.P.); dr.lazareva@bk.ru (E.E.); margarita93a@yandex.ru (M.A.); eblagovechensky@hse.ru (E.B.); 2Centre for Cognition and Decision Making, Department of Psychology, National Research University Higher School of Economics, 101000 Moscow, Russia; k.dzerassa.v@gmail.com (D.K.); kopytin.kg@yandex.ru (G.K.); moiseenko12olesya@gmail.com (O.M.); dobredikhin@edu.hse.ru (D.B.); bermudezmargaretto@gmail.com (B.B.-M.); iannisntoumanis@yahoo.com (I.N.); a.shestakova@hse.ru (A.N.S.); iyaskelaynen@hse.ru (I.P.J.); 3Brain and Mind Laboratory, Department of Neuroscience and Biomedical Engineering, School of Science, Aalto University, 02150 Espoo, Finland

**Keywords:** cognitive function, arthrogryposis, obstetrics palsy, children, motor disorder

## Abstract

The current study aimed to compare differences in the cognitive development of children with and without upper limb motor disorders. The study involved 89 children from 3 to 15 years old; 57 children with similar upper limb motor disorders and 32 healthy children. Our results showed that motor disorders could impair cognitive functions, especially memory. In particular, we found that children between 8 and 11 years old with upper limb disorders differed significantly from their healthy peers in both auditory and visual memory scales. These results can be explained by the fact that the development of cognitive functions depends on the normal development of motor skills, and the developmental delay of motor skills affects cognitive functions. Correlation analysis did not reveal any significant relationship between other cognitive functions (attention, thinking, intelligence) and motor function. Altogether, these findings point to the need to adapt general habilitation programs for children with motor disorders, considering the cognitive impairment during their development. The evaluation of children with motor impairment is often limited to their motor dysfunction, leaving their cognitive development neglected. The current study showed the importance of cognitive issues for these children. Moreover, early intervention, particularly focused on memory, can prevent some of the accompanying difficulties in learning and daily life functioning of children with movement disorders.

## 1. Introduction

The relationship between cognitive and motor development has long been the focus of psychology and psychophysiology. Nonetheless, no consensus has been reached regarding the connection between motor and cognitive skills [1,2,3,4]. Moreover, the neural mechanisms underlying this link remain elusive. Studies have shown significant correlations between particular categories of motor and cognitive function, including complex motor skills and higher-order cognitive abilities. For example, Stöckel and Hughes [5] showed a strong association between anticipatory motor planning and working memory. This motor–cognitive interaction during development was also shown in Abdelkarim’s study [6], which reported that fostering children’s physical fitness during primary school age could enhance both motor and cognitive learning abilities related to academic achievement. However, the authors of one of the most recent reviews on this topic [1] report no significant correlation between motor and cognitive skills in 4 to 16-year-old healthy children. Other studies have revealed that middle school children show a stronger relationship between the main categories of motor and cognitive skills compared to older school children [1].

Importantly, the evidence for a strong connection between cognitive and motor development is not limited to the behavioral domain. Several neurobiological studies have shown co-activation among the cerebellum, basal ganglia, and prefrontal cortex during different motor and cognitive tasks, especially those that are complex, new, have changeable conditions, or require quick responses and concentration to be completed [7,8]. Furthermore, functional studies using fMRI showed the activation of the cerebellum during cognitive tasks in which no movement is involved [9]. A number of anatomical and functional imaging studies have shown that cerebellum function is affected in several cognitive and behavioral developmental disorders, such as attention deficit hyperactivity disorder, autism, and schizophrenia [7]. Importantly, this neurophysiological evidence corroborates the association between motor and cognitive development in children [10].

Thus, cognitive and motor development seem to be particularly interrelated. Indeed, motor and cognitive skills show a similar developmental timeline, especially between the ages of 5 and 10 years old [11], and share several main psychophysiological processes, such as sequencing, monitoring, and planning [12].

Considering the inconsistencies among previous findings, it is crucial to explicitly assess the connection between motor and cognitive development beyond correlational indices. In this sense, the study of clinical populations, in particular children with motor impairment, would significantly contribute to the understanding of the interplay between motor and cognitive function during development. Moreover, the establishment of a clear link between the development of motor and cognitive abilities as well as its underlying brain mechanisms would enable new and integrative rehabilitation approaches for the improvement of both cognitive and motor skills.

To address this question, this study assessed the cognitive function of children with upper limb motor disorders, in particular subjects with arthrogryposis multiplex congenita (AMC) and obstetric brachial plexus palsy (OBPL). Among motor-related diseases, AMC is known to be one of the most serious congenital malformations and is characterized by the presence of two or more major joint contractures, muscle aplasia or hypoplasia, and motoneuronal dysfunction in the anterior horns of the spinal cord. The lack of active movement in the joints of the upper extremities is one of the main problems causing the limitation or inability to self-care. In clinical practice, these motor skills are restored by autotransplantation of muscles from various donor areas. Rehabilitation after such operations is associated with, among other things, neuronal rearrangements in the central nervous system, both in the spinal cord and in the cerebral cortex [13]. OBPL is an injury to the brachial plexus that occurs during birth, usually as a result of a stretching injury from a difficult vaginal delivery. This results in paralysis of the upper limb, which is therefore non-congenital, unlike in the case of AMC [13].

Although the prognosis is generally considered to be good, 20–30% of individuals with OBPL have a residual deficit [14], severe OBPL can result in permanent impairment of arm function, skeletal malformation, cosmetic deformity, behavioral problems, and socioeconomic limitations [15,16]. Individuals with OBPL reportedly have defective motor programming [17]. For example, OBPL infants “forget their arm” during automatic movements [16], supporting the concept of impaired central motor programs in OBPL. Differences in automatic movements between the affected and unaffected sides are caused by incomplete central program development and may contribute to incomplete arm function recovery following OBPL [16]. To further contribute to the discussion of the intertwining of cognitive and motor development, we specifically assessed the state of different cognitive functions, such as attention/concentration, memory, and intelligence, in children with AMC and OBPL as compared with a control group of healthy children. The outcomes obtained in this study may be particularly useful for the development of rehabilitation programs aimed at improving both motor and cognitive skills in children with upper limb motor disorders. Furthermore, these findings confirm the affectation of the central nervous system in this clinical population, as previously suggested in recent studies, thus demonstrating the causal relationship between peripheral motor dysfunction and cognitive impairment [18]. From our point of view, the evaluation of children with motor impairment is often limited to their motor dysfunction, leaving their cognitive development neglected. Therefore, the current study was focused on the importance of cognitive issues for these children.

## 2. Materials and Methods

### 2.1. Participants

A group of 57 children (27 girls) 3–15 years old (mean = 8.3) with upper limb motor disorders (35 subjects with AMC and 22 subjects with OBPL) were selected from the Turner National Medical Research Center for Сhildren’s Orthopedics and Trauma Surgery. A group of 32 healthy children (15 girls) 3–15 years old (mean = 9.6) with no history of visual, hearing, or cognitive disorders were selected as a control group (see Appendix A in Appendix A). Children in the control and motor-impaired groups received the same education according to state standards of general education, thus following the school curriculum for normally developing children.

Regarding their clinical characteristics, children with AMC and OPBL have the following common pathology as per orthopedic classification:

They have the presence of contractures in two or more large joints, hypoplasia or aplasia of muscles, and signs of problems with motoneurons in the anterior horns of the spinal cord. At the same time, the upper limb of a patient has a characteristic profile with the following characteristics: an adductor contracture in the shoulder joint, an extensor (less often, flexion) contracture in the elbow joint, a flexion contracture in the wrist joint, flexion contractures in the fingers, an adduction contracture of the thumb, hypoplasia or aplasia of the muscle of the upper limbs, and restriction or lack of self-service. The muscles of the upper limbs are hypoplastic or absent. Therefore, both AMC and OBPL were included in our research as the clinical group. All patients had symptoms associated with diagnosed diseases and other disorders (for example, brain damage) were not identified.

Children with upper motor disorders were then split into three groups according to their age: Group A (22 children, 3–7 years old), Group B (24 children, 8–10 years old), and Group C (11 children, 11–15 years old) (see Appendix A in Appendix A). Groups were determined in accordance with the most generally accepted age subdivisions in developmental psychology and Elkonin’s periodization, and in correspondence with the three main developmental periods—that is, preschool, primary school, and secondary school age. The same age subdivision was applied to children in the control group, resulting in Group A (6 children), Group B (13 children), and Group C (13 children).

### 2.2. Assessment of Cognitive Functions

A battery of diagnostic techniques was selected to assess children’s cognitive functions of attention span, auditory memory, visual memory, conceptual development, and intelligence. Assessments were conducted individually with each child in a quiet room specially prepared for psychological testing. Two psychologists participated in the evaluation and interpretation of the results.

Attention and auditory working memory were assessed using the Wechsler Intelligence Scale for Children. The WISC-IV was used for children from six years and WPPSI was used for the 3–6-year age group [19,20]. Selected subtests consisted of the repetition of a set of numbers in forward and backward order. The child repeated after the experimenter a set of numbers, first in forward order, then in reverse order, with the opportunity to make one mistake. The total number of memorized digits was recorded on the form and then converted into points, by which the level of attention and memory development was determined.

Attention is the behavioral and cognitive process of selective concentration on a discrete stimulus [21]. In this case, the attention is reflected in the child’s ability to repeat backward the numbers he or she has heard.

Auditory working memory reflects an individual’s ability to listen to information presented orally, encode it, immediately repeat it, and recall it after [21]. In this case, the number of digits the child has memorized reflects the volume of auditory working memory.

Short-term visual working memory was measured using Shipitsina’s “Psychological diagnostics of deviations in the development of children of primary school age.” [22]. Within the framework of this method, 10 pictures were presented one at a time (one picture per second), after which the participant was asked to recall and name the objects presented in the pictures. Visual working memory is a cognitive system that maintains a limited amount of visual information so that it can be quickly accessed to serve the needs of an ongoing task. The number of memorized pictures was recorded on the form and then translated into points, which determined the level of visual short-term memory.

Verbal logical thinking, an aspect of conceptual development that includes generalization processes and the ability to highlight essential features, was measured using the set of sequential pictures of Shipitsina’s “Psychological diagnostics of deviations in the development of children of primary school age.” [22]. After a randomly arranged set of pictures was displayed, children were required to put the pictures in order, making up a logical story. The complexity of the pictures was a function of the age of the participant: the higher the age, the greater the complexity. Thinking was assessed on a point scale, where important evaluation factors were the child’s ability to identify cause-and-effect connections and ways of verbally conveying these connections (number of sentences, number of parts of speech used for this purpose, etc.).

These two tests from the set of Shipitsina were published by decision of the Scientific Council of the Institute of Special Pedagogy and Psychology at the R. Vollenberg International University for Family and Children.

The test for visual memory is very similar to the test VISMEM: Recovery Test (Tombaugh, T. N (1996). Test of memory retention: the TOMM. North Tonawanda, NY: Multi-Health Systems.) Furthermore, the test we have used is more appropriate for our age group.

The test for verbal logical thinking is similar to one of the sets of the Wechsler test (Wechsler, D. Wechsler Intelligence Scale for Children-IV Conceptual and Interpretive Guide. Indiana Univ.-Purdue Univ. Indianap. 2003.), but we did not take it exactly, again, because of the age range, which limited our choice of methods.

Finally, intelligence was evaluated using Raven’s progressive matrices (A, B, C) [23]. We used two types of tests-CPM/CVS kit and SPM+/MHV, because of the age period of the children group. All children in group A are very close to four-year-old, therefore they were included into group A (older than three years and seven months). Participants were shown a series of pictures with progressive patterns and asked to choose the piece that logically fit the picture. Intelligence is the ability to think, learn from experience, solve problems, and adapt to new situations. The number of correct answers was converted to a point system adapted for each age period.

The tests containing this assessment battery were chosen partly based on the age range of the sample (from 3 to 15 years old) and in consideration of the time constraints for the diagnosis of each child (that is, 60 min, due to medical reasons). These methods thus allowed us to fully assess the main cognitive functions of motor-impaired and healthy children representing a wide range of ages.

General motor development (GMD) refers to a person’s functional abilities, which were evaluated by a neurologist based on the ball system of self-care skills. The maximum number of points is 21. Subjects with a mild extent of functional impairment (level 3) scored 17–20 points on the self-care scale ovals. A moderate extent of severity (level 2) corresponds to 9–16 points on the self-care scale, and a severe extent (level 1) corresponds to a self-care score of 8 or lower.

### 2.3. Statistical Analysis

All cognitive tests that were used to assess the performance of the participants provided balanced, non-skewed scores. Accordingly, we treated the participants’ scores as interval data, which allowed us to evaluate the results by means of quantitative analysis. Specifically, the modulation of cognitive performance by the factors of group (either subjects with motor impairment or control children) and age (younger, medium, and older children) was tested by the means of a series of 2 × 3 univariate analyses of variance (ANOVA). To account for the family-wise error rate, the resulting *p*-values were corrected with respect to the false discovery rate (FDR) according to Benjamini and Hochberg [24], with a critical *q*-value of 0.15.

Further, the link between GMD and performance in each cognitive task was evaluated by means of Kerndall’s correlation coefficient (Kendall’s tau), which is recommended by Khamis [24] for data sets of our type. Similar to the series of univariate ANOVAs, the revealed *p*-values were corrected with respect to the FDR according to Benjamini and Hochberg [25], with a critical *q*-value of 0.15.

## 3. Results

The ANOVA showed that nearly all cognitive indices obtained across different tests were significantly affected by the age of the participants (see Table 1). In particular, older children were characterized by higher scores in attention, auditory and visual memory, storytelling, and average cognitive score (ACS) (see Appendix A in Appendix A).

Importantly, results revealed that scores obtained in both auditory and visual memory tests, the verbal-logical aspect of thinking, and the ACS were significantly different between the two groups (see Appendix A in Appendix A). Specifically, across all these tests, the performance of subjects with motor impairments was significantly lower than that of the control group (see Figure 1 for the score distributions in auditory and visual memory tasks).

Further, we examined the interaction of GMD of the subjects and their performance in various cognitive tasks. As can be seen in Table 2, these analyses showed that auditory memory was significantly correlated with the GMD of children (*r**_τ_* = 0.26; *p* = 0.02, *q* = 0.13), as well as the attention span (*r**_τ_* = 0.24; *p* = 0.04, *q* = 0.11). Moreover, we also observed a relation between participant’s visual memory and their GMD as a trend (*r**_τ_* = 0.20; *p* = 0.08; *q* = 0.16). Therefore, although we do technically reject the hypothesis regarding the correlation between participants’ motor function and visual memory performance, we consider that a particular link between those variables might still be possible as a trend. Performances on the other tests were not significantly explained by GMD scores.

## 4. Discussion

In the present study, we analyzed the effects of upper limb motor disorders on the development of cognitive functions as assessed with neuropsychological tests. Our results showed the difference in cognitive performance between subjects with motor disorders and age-matched healthy controls. Specifically, we report a significant effect of motor impairment on memory performance in both auditory and visual domains, as well as in thinking and ACS. At the same time, performance on the tests evaluating attention, intelligence, and storytelling did not differ between children with upper limb motor disorders and their peers in the control group. Overall, we emphasize that ACS was significantly decreased in children with upper limb motor disorders.

We compared the performance of cognitive tasks between children with upper limb motor disorders and healthy controls across three age groups: preschool, primary school, and secondary school. We assessed the span of auditory and visual memory, attention, intelligence, VLT, and ACS. We also ran a separate regression analysis of GMD and subjects’ cognitive development.

A separate analysis of different age groups allowed us to trace cognitive development in children with upper limb motor disorders and control children dynamically. As a result, we observed the delay of cognitive development in children with upper limb motor disorders on their memory performance in both visual and auditory domains. Specifically, the difference in memory performance between subjects with upper limb motor disorders and control children was most prominent in the range of 8–10 years, whereas older children caught up to their control peers.

It is noteworthy that in this age range, cognitive learning styles and personality are actively formed, which allows the child to form their own individual behavioral programs. Overall, this age is referred to as the period of active formation of voluntary regulation of behavior, reflection, and self-control [26]. Basically, children of this age are actively growing, and their brains are developing intensively. At a deeper level, subtler functional connections are being formed between different brain areas, which will ensure the complex work of the whole organism [27]. It has also been shown that children 8–10 years of age undergo the accelerated formation of those brain areas that are responsible for motor activity. Accordingly, their movements become more accurate and varied [28,29]. Our results show that these processes might not fully occur in children with motor development disorders at this age and essentially are shifted to an older age. Most importantly, our findings reveal that such motor impairment is likely responsible for the cognitive delay shown in these children, particularly with respect to visual and auditory memory.

Previous studies have shown that motor and cognitive development can be fundamentally attributed to ages 8–10 [12]. Moreover, contrary to the previously widespread belief that motor development begins and ends earlier than cognitive development, studies have shown that both motor and cognitive development have equally long and likely interconnected developmental schedules [29,30,31]. Considering the present findings, a particular connection between cognitive and motor development seems plausible. Specifically, impaired cognitive development (e.g., in the case of a mental disorder) is likely to lead to impaired motor development. However, in the case of children with upper limb motor disorders, an inverse relationship might be assumed: if motor development is delayed, the development of cognitive functions (especially memory) seems to be affected too. Studies suggest that some aspects of cognitive and motor control are highly correlated, especially in the age range of 8–10 years old [12,32]. The difference in memory performance between children with upper limb motor disorders and healthy children, as well as the significantly different VLT performance between these groups, also implies such a connection.

To reveal this motor–cognitive link in detail, we further attempted to explicitly analyze the association between motor development in children with upper limb motor disorders and their cognitive development assessed by means of a wide battery of tests. Importantly, we found a significant correlation between subjects’ GMD and auditory memory performance. Moreover, we observed a correlation between the visual memory and the GMD at the level of the trend. However, we observed a correlation between the visual memory and the GMD at the level of the trend. Moreover, we suggest that the significant correlation between attention and general motor development might also be connected to the processes underlying decreased memory performance due to the strong link between attention and memory [33]. Of note, had we only focused on memory performance as the main aspect of cognitive development affected by impaired motor development, we would have found a significant connection between motor development and both modalities of memory. Accordingly, although we could not find a significant correlation between visual memory performance and cognitive development, a particular link between them might still persist. At the same time, our findings with regard to auditory memory are fully in line with previous studies showing the link between memory and motor skills [27,32,34]. We also suggest that GMD might be connected with other modalities of memory during a particular age range, and hence its effect could be blurred since the correlation analyses in the current study could not be performed separately for specific age ranges but rather were performed for the whole group of children. Future studies might attempt to clarify the link between the motor development of children and their memory function in more detail; in this vein, we suggest using a larger sample size, which would allow us to conduct separate correlation analyses of different age groups.

We found no significant correlation between motor development and intelligence. The absence of such a relationship is in line with other studies confirming that motor and intellectual levels in healthy children are largely independent [35]. Likewise, no correlation was observed between motor development and attention, which is also in line with previous studies [30]. Moreover, VLT did not correlate with motor development in children with upper limb motor disorders, despite the fact that we observed a significant difference in VLT performance between this group and the control group. We suggest that the lack of such correlation might be because VLT is an indicator of a broad range of cognitive functions, not limited to memory. VLT begins to develop when more and more words are memorized, speech is embedded in children’s activities, and they start to perform a planning function. Thus, VLT performance might be dependent on memory performance, but not determined by it, as it is also affected by other domains of cognitive functions.

Summing up, our results indicate that a connection between children’s motor development and memory performance does exist, at least in the auditory domain. In order to account for a possible explanation for that phenomenon, developmental changes in both motor and cognitive function must be considered.

First, children participating in this study were experiencing a period of active improvement of motor skills, coordination, muscle control and reaction time, and facilitation of the coordination of the biggest muscles, which altogether leads to success in organized sports and games, improved coordination of small muscles, the mastery of complex own skills, and improved fine control [36]. Moreover, around the age of 8–10 years old, one might observe a noticeably smoother combination of motor actions and motor skills compared to younger children. Specifically, normal children can rotate, twirl and jump, and perform tasks that help them in sports [37,38]. In the cognitive sphere, this period is referred to as the concrete operational stage, a term proposed by the Swiss psychologist Jean Piaget. The concrete operational stage refers to the developmental period at which children begin to apply logic and goals to specific events [39]. It is also the time when an infant’s brain undergoes a series of significant changes. Information passes through the nervous system at a faster rate, and different parts of the brain begin to work in concert with each other in new combinations [40].

We suggest that the delay in the cognitive development of children with upper limb motor disorders, particularly observed in memory performance, might occur because the simultaneous development of the aforementioned motor skills, which is necessary during the concrete operational stage, does not occur in children with upper limb motor disorders. Moreover, motor and cognitive functions might be connected on a deeper level, and hence may depend on the development of the same cortical and subcortical structures [41].

The interconnection of memory and motor development has also been observed in other studies, which have shown the effect of motor memory in action choice and in procedural learning. Thus, motor memory, considered an intrinsic property of the motor system, can impact not only motor behavior, depending on the constraints, but also higher cognitive functions [42]. Indeed, current theories on memory function consider that both declarative knowledge and procedural skills might be acquired based on sensorimotor interaction and interactive behavior [43,44,45].

Altogether, the assessment of cognitive skills in children with impaired motor abilities in the current study contributes to a better understanding of the complex interconnection between motor and cognitive development. Our analysis shows that memory seems to be the primary aspect of cognitive development affected by impaired motor function. For a more detailed understanding of such a link between memory and motor development, a more specific study is required [46].

Regarding the practical implications of this study, these results should be reflected in individualized educational and rehabilitation approaches for children with motor disorders. Whereas special rehabilitation programs designed for children with different disabilities are widely used nowadays [47], children with motor impairments need programs specifically adapted to their needs, considering all the nuances of their cognitive development, as reflected in this study. The enhancement of motor development with, for example, the help of interactive video games may be a new avenue of experimental research. Along these lines, such games could be used to motivate children to slowly train their undeveloped muscles, and whether or not this influences their cognitive development could then be evaluated [48,49,50]. In the future, this approach could help to understand exactly what aspects are required to be considered during the development of rehabilitation and habilitation programs for motor-impaired children.

## 5. Conclusions

Motor dysfunction of the upper limb does impair cognitive functions, especially auditory and visual memory.The link between cognitive skills and motor impairment is especially manifested between the ages of 8 and 10 years old.These findings must be considered in the development of rehabilitation programs for individuals with motor disorders.

## Figures and Tables

**Figure 1 brainsci-11-01650-f001:**
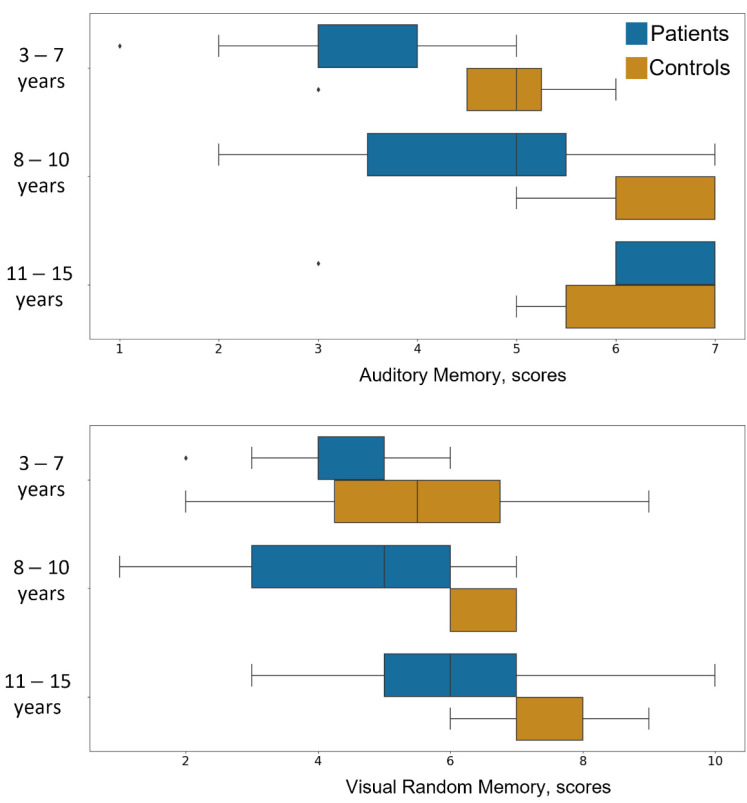
Scores in memory-assessment scales for patients (blue) and control children group (orange). Upper panel: Auditory Memory performance; lower panel: Visual Memory performance. Control group is characterized with higher memory performance despite the age of participants. ◆ - points outside of 1.5 interquartile range (IQR), which were identified as outliers.

**Table 1 brainsci-11-01650-t001:** Statistical results for cognitive performance (ANOVA) as a function of factors Group (df = 1; either patient or control), and Age (df = 2: either 3–7 years, 8-10 years, or 11–15 years).* *p* < 0.05, ** *p* < 0.01, *** *p* < 0.001.

	Group				Age				Interaction			
	F (1,64)	*p*	$\eta_{p}^{2}$	*q*	F (2,64)	*p*	$\;\eta_{p}^{2}$	*q*	F (2.64)	*p*	$\eta_{p}^{2}$	*q*
Attention	2.64	0.11	0.04	0.15	15.63	< 0.001 ***	0.32	< 0.001	0.72	0.49	0.02	0.57
Auditory Memory	11.72	< 0.01 **	0.15	< 0.01	15.01	< 0.001 ***	0.32	< 0.001	0.94	0.4	0.03	0.56
Visual Memory	15.59	< 0.001 ***	0.19	< 0.001	11.07	< 0.001 ***	0.25	< 0.001	1.06	0.35	0.03	0.82
Intelligence	0.32	0.57	0	0.57	0.58	0.57	0.02	0.57	0.64	0.53	0.02	0.53
Storytelling	1.45	0.23	0.02	0.27	4.87	< 0.05 *	0.13	< 0.05	1.75	0.18	0.05	1
Thinking	16.55	< 0.001 ***	0.2	< 0.001	1.34	0.27	0.04	0.31	1.09	0.34	0.03	1
ACS	18.51	< 0.001 ***	0.22	< 0.001	22.19	< 0.001 ***	0.41	< 0.001	1.02	0.37	0.03	0.64

**Table 2 brainsci-11-01650-t002:** Correlation analysis (Kendall’s tau) between the general motor development (GMD) of patients and their performance across the different cognitive tasks. * *p* ≤ 0.05 and satisfies the FDR control (*q* ≤ 0.15).

	$r_{\tau }$	*p*	*q*
Attention	0.24	0.04 *	0.11
Auditory Memory	0.26	0.02 *	0.13
Visual Memory	0.2	0.08	0.16
Intelligence	−0.05	0.66	0.8
Storytelling	0.04	0.76	0.76
Thinking	−0.09	0.42	0.63

## Data Availability

The data presented in this study are available on request from the corresponding author. The data are not publicly available due to patient confidentiality.

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
