# Peer review of "Aberrant Auditory and Visual Memory Development of Children with Upper Limb Motor Disorders"

_brainsci, 2021, doi:10.3390/brainsci11121650_

Round 1
Reviewer 1 Report
1.The result was that hearing and visual memory were impaired at the age of 8-10 when there was an upper limb disorder, but it caught up with the older age. Why catch up? If they catch up, we don't need any intervention. When tracking the same person, does it show that even if the memory is impaired at the age of 8-10, it will catch up with the older age?
2.Was there a lack of quality or quantity of rehabilitation among the children in the upper limb disability group?
3.Did the upper limb disorder group really have no brain damage originally? Is it possible that there was some damage to the brain if it was paralysis of labor? Is head MRI examination done in advance?
Author Response
We wish to very much thank you for the important comments and advice. Please see the attachment.

Reviewer 2 Report
This is an interesting article and has implications for understanding of the development of memory as well as implications for early intervention and education.
The introduction is well written. I have quite a few questions about design and analysis. I'm not sure that my questions about selection of tests can be addressed. I couldn't see that any of the tests were suitable for the younger participants. One of the tests isn't published in English, so I couldn't be clear, but the other tests are well known and there is no reliability or validity data for use with younger children. I also have queries about the statistical analysis. My main comments are as follows:
I didn't see any information on recruitment of the 'control' participants. It is noted that the participants were age matched, but it wasn't clear how that occurred. It would be helpful to have the mean and range for age for each group and the gender. This could be in a table.
It wasn't clear why the numbers of participants in the control and GMD groups were different. In Group A there is more in GMD (22) than control (6), for Group B there is 23 GMD and 13 control, for Group C there is 11 GMD and 13 control. I couldn't see the logic for these numbers.
Line 121 indicates the age groupings were 3-6, 7-10 and 11-15. Figure 1 has the grouping as 3-7, 8-10, 11-15. There is an error somewhere.
WISC subtests were used to assess working memory. The WISC-IV is for children 6+ years. I wondered why the working memory subtests from the WPPSI weren't used for the 3-6 year age group. These are the correct tests to use for this age group.
Raven's Progressive Matrices were also used and my understanding is the starting age for RPM is 4 years.
The "Shipitsyna, L.M. Psychological diagnosis of deviations in the development of children of primary school" was used. I couldn't find any information in English but it also looks like a test for older children.
It would be helpful to indicate which version of the WISC was used. Was it the WISC-V, published in 2016? Reference 19 is for the WISC-IV.
Line 83 "with OBPL have a residual deficit [14]. Severe OBPL can result in permanent impairment" - I think there should be a comma rather than a full stop i.e. with OBPL have a residual deficit [14], severe OBPL can result in permanent impairment...
I didn’t see any clear hypotheses or research questions. The first hypothesis I saw was mentioned in the results section relating to the findings from the linear regression.
I found the results section difficult to read and perhaps it could be handled more efficiently. It would be helpful to start with a table of descriptives including the means and standard deviations for all measures for the GMD and control by age group. This could replace Figure 1. It would require the same space but would be more informative.
I understand that the age groups map onto educational classifications. Developmentally though, the groupings don’t make as much sense. The difference between a 3 year old and a 6 year old is much greater than the difference between a 7 year old and a 10 year old, for example. This creates problems for use of these age categories as interval data. With interval data, there is an assumption of equal distance between each point. My suggestion is to run the ANOVA again and use age as a continuous variable. This will be more powerful and more accurate. I would also recommend adding gender to the equation.
The ANOVAs are not reported correctly. Results from https://www.mdpi.com/2076-3425/11/7/935/htm and https://www.mdpi.com/2076-3425/9/12/356/htm are both reported in Brain Sciences and show typical reporting of an ANOVA with inclusion of the effect size. It is important to follow the conventions of reporting to support the reader and for any systematic reviews/meta-analyses that might be conducted in the future.
The regression analysis is listed as ‘ordinary’ – but I think it should be ‘ordinal’. The regression analysis seems to be used as a correlation. I wasn’t sure what the regression contributed. Also, given the average age of the participants in the control condition is likely to be substantially higher than for the GMD group, correlations/regressions without correction for age are not meaningful.
This isn’t a clinical intervention, so the participants shouldn’t be referred to as patients. I would also recommend avoiding ‘subjects’ as it is declining in use. ‘Children’ or ‘participants’ would be preferable and consistent with guidelines such as those published by the American Psychological Association.
To include age as a continuous variable, you will need to run an Analysis of Covariance (ANCOVA) with age as the Covariate. This will allow you to statistically control for age. In SPSS, go to General Linear Model, then select Univariate. The output will provide all the information you need for correct reporting of ANOVA.
I hope these comments are helpful.
Author Response
We wish to very much thank you for the important comments and advice! Please see the attachment.

Round 2
Reviewer 1 Report
The authors politely answered all the questions.
Author Response
Thank you one more time for your valuable comments and advice!
Reviewer 2 Report
Thank you for the comprehensive corrections. I am particularly relieved that the correct tests were used for the younger participants. It would have been a major threat to the validity of the study if this wasn't the case.
I've checked through and everything requested has been addressed and I have no further questions. I only ask for one very minor change which I'm sure can be handled between authors and editor. It would be helpful to make reference to the tables added in the supplementary materials within the manuscript. Perhaps reference to the results for the descriptive statistics could be added to the first paragraph of the results and reference to the ANCOVA could be added after the results for the ANOVA.
I appreciate the clear responses to the issues raised. It made the task of reviewing much easier.
Author Response
We have added links to the latest version of the paper, as per your recommendations (Line 111,129,222,227).
Thank you one more time for your valuable comments and advice!